# Horse Innate Immunity in the Control of Equine Infectious Anemia Virus Infection: A Preliminary Study

**DOI:** 10.3390/v16121804

**Published:** 2024-11-21

**Authors:** Giusy Cardeti, Giuseppe Manna, Antonella Cersini, Roberto Nardini, Sergio Rosati, Ramses Reina, Marina Cittadini, Stefania Sittinieri, Alessia Altigeri, Gaetana Anita Marcario, Maria Teresa Scicluna

**Affiliations:** 1Istituto Zooprofilattico Sperimentale del Lazio e della Toscana “M. Aleandri”, Via Appia Nuova 1411, 00178 Rome, Italy; giusy.cardeti@izslt.it (G.C.); giuseppe.manna@izslt.it (G.M.); antonella.cersini@izslt.it (A.C.); marina.cittadini@izslt.it (M.C.); stefania.sittinieri@izslt.it (S.S.); alessia.altigeri@izslt.it (A.A.); gaetanaanita.marcario@izslt.it (G.A.M.); teresa.scicluna@izslt.it (M.T.S.); 2Department of Veterinary Science, University of Turin, Largo P. Braccini 2, 10095 Torino, Italy; sergio.rosati@unito.it; 3Instituto de Agrobiotecnología IDAB-CSIC, Av. de Pamplona, 123, Mutilva Baja, 31192 Navarra, Spain; ramses.reina@csic.es

**Keywords:** equine infectious anemia virus, horse, innate immunity, macrophage, cytokine, gene expression

## Abstract

The mechanisms of the innate immunity control of equine infectious anemia virus in horses are not yet widely described. Equine monocytes isolated from the peripheral blood of three Equine infectious anemia (EIA) seronegative horses were differentiated in vitro into macrophages that gave rise to mixed cell populations morphologically referable to M1 and M2 phenotypes. The addition of two equine recombinant cytokines and two EIA virus reference strains, Miami and Wyoming, induced a more specific cell differentiation, and as for other species, IFNγ and IL4 stimulation polarized horse macrophages respectively towards the M1 and the M2 phenotypes. Infection with EIAV reference strains resulted in a morphological transformation of macrophages compatible with the M1 differentiation pattern. All samples were also analyzed by molecular analyses for equine herpesviruses that could have acted as an interference and were found to be negative. The mRNA expression level of the pro-inflammatory genes *MMP13* and *IL6* in treated equine monocyte-derived macrophages (eMDMs) was evaluated by a SYBR^®^ Green real-time PCR. In this study, *MMP13* represented a reliable target gene to evaluate pro-inflammatory status of macrophages in horses because IFNγ and EIAV infection considerably increased its expression. A more in-depth study of the expression genes of both cytokine-induced and virus-induced markers of eMDM polarization may help us to understand whether these markers in horses are the same as those found in other animal species with similar pathways of innate immunity activation. The identified markers of each macrophage population would allow analysis of the differentiation profiles that could provide information on virus infectivity control in equid populations, envisioning their use in therapeutic strategies.

## 1. Introduction

Innate immunity is the first line of defense against pathogens and enables the control of lentivirus infections in animal species, such as those in primates, cats, and small ruminants [1]. Equine infectious anemia virus (EIAV), a member of the genus Lentivirus, family Retroviridae, induces an acute, chronic, or inapparent clinical disease in horses, with the latter clinical form predominant in donkeys and mules [2]. The modest blood viral titers observed in animals in which infection takes an asymptomatic course (non-progressors), rule out that infection control is based solely on high antibody titers induced by the high viral loads that is associated with acute phases of the infection [3]. As reported in the literature, following recognition of the pathogen, the transduction signals of immune cells that characterize innate immunity (macrophages and/or dendritic cells) induce a series of events that can counteract infection, inhibiting in some cases the viral replication cycle [1]. Macrophages display high phenotypic and functional heterogeneity with a spectrum ranging from classical pro-inflammatory (M1) macrophages to reparative (M2) macrophages [4]. Alterations in synovial macrophage functional activity is implicated in the pathogenesis of lentiviral infections [1,5]. However, these mechanisms of viral control in EIAV-infected non-progressors are not yet widely described.

The purpose of this study was to investigate the macrophage polarization and to assess the expression of genes related to the horse innate immune system following stimulation with cytokines and in vitro infection with EIAV.

## 2. Materials and Methods

Monocyte/macrophage isolation was obtained from peripheral blood that was collected at the slaughterhouse from one 7-year-old male pony and two Arabian female horses, respectively 9 and 10 years old. All the animals tested seronegative for EIAV by ELISA [6], AGID (210421_EIA_AGID_SOP_JCv2) [7], and immunoblotting [8] serological techniques in use at the National Reference Center for Equine Infectious Anemia and WOAH Reference Laboratory, IZSLT, Rome, Italy.

### 2.1. PBMC Separation from Equine Whole Blood

The protocol developed required the collection of at least 12 mL of ethylenediaminetetraacetic acid (EDTA) blood. After a clarification at 115× *g* for 4 min, leucocytes were collected along with plasma and transferred in a new tube and centrifuged at 1500× *g* for 15 min.

Plasma was discarded and the pellet was resuspended in 4 mL of phosphate buffer solution (PBS)-0.004 mM EDTA. This suspension was carefully layered onto 4 mL of Histopaque^®^ (Sigma Aldrich, Darmstadt, Germany). After a centrifugation at 400× *g* for 30 min, the supernatant was discarded and peripheral blood monocyte cells (PBMC) were collected and resuspended in 10 mL of PBS-EDTA. The suspension then underwent two cycles of centrifugation at 800× *g* for 10 min, and after the second cycle the pellet was resuspended in 10 mL of cell culture medium (T-I: RPMI medium added with 10 mM sodium pyruvate, 100 UI/mL penicillin, 100 ug/mL streptomycin, 2 ug/mL amphotericin B, 1 mg/mL gentamicin, 0.2 M L-glutamine, 50 µM 2-mercaptoethanol).

### 2.2. Equine Monocyte-Derived Macrophages (eMDMs) Isolation and Stimulation

A protocol for isolation and stimulation of eMDMs with equine cytokines or with EIAV was adapted from Raabe et al. (1998) [9] and Ma et al. (2014) [10].

At each of the three different blood collections, about 12 mL of EDTA-blood sample was obtained. After separation, PBMCs were counted using a hematological analyzer to adjust their concentration to obtain 7.5 × 10^5^ cells/mL and were thereafter dispensed at a volume of 9.6 mL in equal parts of 0.8 mL in a 12-well cell culture microplate and left to adhere on the well surface. Following 18–20 h of incubation at 37 °C and at 5% CO_2_, non-adherent and loosely adherent cells were removed by washing quickly for three times with cold PBS and with the subsequent addition of fresh complete medium T-II (T-I supplemented with 10% heat-inactivated horse serum). On the third day, most of the adherent cells differentiated into macrophages and were either stimulated with equine recombinant cytokines IFNγ (Abcam, Waltham, MA, USA) or IL4 (Abcam, Waltham, MA, USA) at 50 ng/mL in T-II medium, or infected [11] with Miami (IBVR VIR RE RSCIC 57) or Wyoming (IBVR VIR RE RSCIC 110) reference strains of EIAV (purchased from IZSLER Biobanking of Veterinary Resources, IBVR) at a multiplicity of infection (MOI) = 0.1. The cytokine treatment lasted for 7 days, whereupon some of the stimulated eMDMs were infected with either one of the two EIAV reference strains on day 10. Therefore the supernatant of the differently treated eMDM cultures was collected weekly for 1 month, starting from the day of infection.

After cytokine treatment, non-infected eMDMs were collected by scraping the wells making sure that all the cells were hoarded, to evaluate the mRNA expression level of the pro-inflammatory genes matrix metalloproteinase 13 (*MMP13*) and *IL6*. The same study analyses were performed on untreated eMDMs in two wells, that were each infected with one of the two reference viral strains and where the supernatants were collected at 7 and 10 days post-infection (PID).

### 2.3. Molecular Analyses

Molecular analyses were conducted on 3 blood and 29 supernatant samples, collected weekly from the 3 eMDM cultures stimulated with cytokines and infected with EIAV as also from those only infected. The 3 blood samples were examined for EIAV by real-time TAT PCR [12] and nested LTR-TAT PCR [13], both from DNA and RNA templates. Varicellovirus equidalpha 1 and 4 (EqHV1/4) [14], Percavirus equidgamma 2 and 5 (EqHV-2 and EqHV-5) [15] were also targeted. In addition, the supernatants of eMDM cultures were analyzed using a pan-herpesvirus PCR [16] for the presence of the previously mentioned equine herpesviruses that could initially integrate into the cellular genome and replicate while the macrophages are in culture.

The DNA and RNA nucleic acid extractions were both carried out using the QIAsymphony (QIAGEN Gmbh, Hilden, Germany) with the QIAamp^®^ Viral RNA Mini kit (QIAGEN Gmbh, Hilden, Germany). The sensitivity for the detection of the EIAV from an RNA template, was improved by carrying out a cDNA synthesis using 20 µL of RNA with the High-Capacity cDNA Reverse Transcription Kit (Applied Biosystems, ThermoFisher Scientific, Waltham, MA, USA), following the manufacturer’s instructions and using the Gene Amp^®^ PCR System 9700 (Applied Biosystems, ThermoFisher Scientific, Vilnius, Lithuania) with a thermal cycling profile of 10 min at 25 °C and 45 min at 37 °C.

A SYBR^®^ Green real-time PCR was used to evaluate the mRNA expression level of the pro-inflammatory genes *MMP13* and *IL6* in eMDMs treated with either IFNγ, IL4 or infected with either one of the two EIAV reference strains [11].

The nucleic acid extractions (both DNA and RNA) were carried out using the QIAsymphony (QIAGEN) with the QIAamp^®^ Viral RNA Mini kit (QIAGEN).

Three separate in-house SybrGreen PCRs were performed using three pairs of primers eMMP13, eIL6, e18S, the HOTFIREPol^®^ Multiplex qPCR Mix (ROX) (Solis BioDyne OÜ, Tartu, Estonia), and the fluorescent dye Eva Green (Biotium, Fremont, CA, USA). The amplification thermal profile used was 95 °C for 10 min, 40 cycles at 95 °C for 15 s, 60 °C for 60 s. For the HRM analysis, the thermal profile used was the following: 95 °C for 15 s and continuous heating from 60 °C to 95 °C with a ramp of 0.05 °C/s. At the end of the PCR runs, the amplicons and melting curves obtained for each of the markers were compared to those produced by the samples.

Four replicates of the following six eMDM samples were analyzed: only stimulated for 7 days with IFNγ or IL4 and only infected with either EIAV Wyoming or Miami strain, at 7 and 10 PID [11].

## 3. Results

### 3.1. Morphology of eMDM Following Isolation and Stimulation

The eMDMs were isolated from blood collected from three serologically EIAV negative horses. Four days after the treatment/infection, the shape and size of the macrophages started to change. Based on the morphology, stimulation with IFNγ polarized the macrophages towards an M1-type (macrophage with inflammatory profile) morphology of differentiation that was similar in shape to a more or less irregularly shaped fried egg, in contrast to stimulation with IL4, which induced M2-type shaped cells (macrophage with homeostatic functions) characterized by the emission of long pseudopodia, similar to fibroblastic cells (Figure 1). Cell death for M1 was observed three weeks from stimulation with IFNγ, while cell death for M2 was detected three or four weeks from stimulation with IL4.

The majority of the macrophages infected with the EIAV strains showed an M1-like morphology. Interestingly, M1 eMDMs infected with the EIAV died before M2 eMDMs, that was about seven days from inoculation with the two EIAV reference strains.

### 3.2. Molecular Analyses

The EIAV genome was not detected in the three equine whole blood samples by real-time TAT PCR or by nested LTR-TAT PCR using both DNA and RNA templates. All samples tested for equine herpesviruses tested negative.

The mRNA expression level of the pro-inflammatory genes *MMP13* and *IL6* in eMDMs treated with IFN-γ, IL4 were generally low in cytokine-stimulated eMDMs, while higher expression levels were observed in macrophages infected with EIAV reference strains.

Comparing the relative expression of pro-inflammatory markers *MMP13* and *IL6*, the levels of the former were up approximately six times higher than the levels of *IL6* (Figure 2).

In particular, *MMP13* was overexpressed when compared to *IL6* in the following cases: stimulation with IFNγ (approximately 6 times), 10 days after infection with the EIAV Wyoming strain (approximately 4 times), 7 days after infection with the Miami strain (approximately two and a half times), and 7 days after infection with the EIAV Wyoming (approximately one and a half times). In contrast, 10 days after the infection with the Miami strain, the two markers were expressed in an equivalent manner.

## 4. Discussion

According to the results of this preliminary study, horse monocytes from peripheral blood can be differentiated in vitro into macrophages, which generally gives rise to mixed populations morphologically referable to M1 and M2 phenotypes. The addition of a particular cytokine, or a viral infection, results in a more specific differentiation. As stated for other species [1], stimulation with equine recombinant cytokine IFNγ polarized horse macrophages towards the M1 phenotype, while equine recombinant cytokine IL4 polarized them towards the M2 phenotype. Viral infection (EIAV Wyoming and Miami strains) resulted in a morphological transformation of macrophages compatible with the M1 pattern of differentiation.

Furthermore, in this study, *MMP13* represented a reliable target gene to evaluate the pro-inflammatory status of macrophages in horses as IFNγ and EIAV infection considerably increased its expression. In fact, *MMP13* is related to a specific inflammatory condition [17] but also is reported as being involved in infectious diseases such as bovine respiratory infection [18].

A more in-depth study of the expression genes of both cytokine-induced and virus-induced markers of eMDM polarization may help us to understand whether these are the same in the horses as those found in other animal species, such as pigs, sheep, goats [1], and humans [19] with similar modes of activation of innate immunity. The identification of the markers of each macrophage population would allow us to analyze the differentiation profiles, and to check the control of virus infectivity in both populations, envisioning therapeutic strategies through the use of this information.

## Figures and Tables

**Figure 1 viruses-16-01804-f001:**
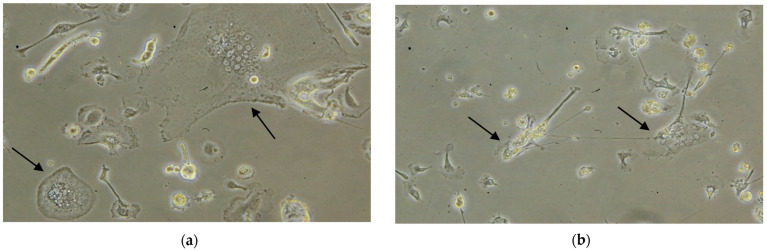
Different shapes and patterns of eMDM depending on the stimulating cytokine. (**a**) M1-shaped eMDM (arrows) after stimulation with IFNγ; (**b**) M2-shaped eMDM (arrows) after stimulation with IL4.

**Figure 2 viruses-16-01804-f002:**
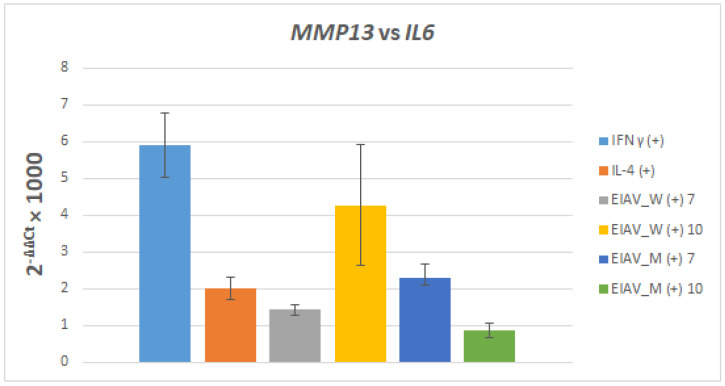
*MMP13* relative to *IL6* mRNA expression in eMDMs from an EIAV-seronegative donor. eMDMs were stimulated (+) with cytokines or with EIAV strains Wyoming (W) or Miami (M) at 7 and 10 days. Units 2^−ΔΔCt^ × 1000.

## Data Availability

The data presented in this study are available upon request from the corresponding author.

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
