# Peer review of "Horse Innate Immunity in the Control of Equine Infectious Anemia Virus Infection: A Preliminary Study"

_viruses, 2024, doi:10.3390/v16121804_

Round 1

Reviewer 1 Report

Comments and Suggestions for Authors

The present study characterizes the polarization of host macrophages upon stimulation with cytokines and in vitro equine infectious anemia virus (EIAV) infection. Identification of gene signatures in M1 and M2 macrophages was assessed using by qPCR. The investigation aims to provide scientific data for further studies on immunity and pathogenic mechanisms of EIAV infection.     

 According to this reviewer, the manuscript is of interest and contains information suitable for a communication. However, it seems that the collected data has not been statistically evaluated. Error bars describing the variation in observed data are not included in the provided figures. I encourage the authors to analyze statistically the data collected from technical and biological replicates. 

Author Response

We try to give a response as best we can to all the comments of the reviewer. Please see the attached Template.

We apologize but it is not possible to respond to all the reviewer's comments because the work was conceived to be published as a short communication and not as a detailed article.

Reviewer 2 Report

Comments and Suggestions for Authors

Equine infectious anemia is one of the important equine diseases.

The present manuscript will provide new information about innate immunity of EIA.

The experiments are designed well, although some experiments should be added to improve the quality of the present manuscript.

Specific comments are as follows:

L25: Add the full name of  eMDM

L49: The authors should provide enough information about the macrophage polarization and EIAV infection in Introduction to help readers.

L83: [Menarim, 2020] should be [14].

L83-84: The full name of IBVR and location, (Isler Biobanking of Veterinary Resources, Brescia, Italy?), should be added. 

L128-141, L188-191: Polarization to M1 or M2 should be confirmed by the detection of markers for each type of macrophages. And also infection of EIAV should be confirmed by the detection of EIAV antigens. It is not difficult to identify cell types using immunofluorescence assay. Double staining for macrophage typing and EIAV infection is recommended to show experimental evidence even though the authors report preliminary results.

L140-141 and L177-179: The authors should show shape and marker detection results using photos.

L140-141; The authors should show data about cell death for M1, M2, EIAV infected cells.

L148-151: The expression levels of MMP13 and IL6 seems to be different between the two reference EIAV. The authors should describe the results and discuss about the difference between the two viruses.

L153-156: What is the units of Y-axis in Figure 2 (a) and (b)?

L157-169: Expressions of MMP13 and IL6 are almost nothing in Figure 2. What is the meaning of comparing the relative expressions of these markers?

Finally, decimal marker should be shown using point on the line in the present manuscript written in English according to British practice.

Author Response

(The authors gave the same response as above.)

Round 2

Reviewer 1 Report

Comments and Suggestions for Authors

The paper is suitable for publication as a communication. 

Reviewer 2 Report

Comments and Suggestions for Authors

The responses to the comments were appropriate.

The manuscript has been revised appropriately.